# Modelling the Effect of MUC1 on Influenza Virus Infection Kinetics and Macrophage Dynamics

**DOI:** 10.3390/v13050850

**Published:** 2021-05-07

**Authors:** Ke Li, Pengxing Cao, James M. McCaw

**Affiliations:** 1School of Mathematics and Statistics, The University of Melbourne, Parkville, VIC 3010, Australia; pengxing.cao@unimelb.edu.au (P.C.); jamesm@unimelb.edu.au (J.M.M.); 2Peter Doherty Institute for Infection and Immunity, The Royal Melbourne Hospital and The University of Melbourne, Parkville, VIC 3010, Australia; 3Melbourne School of Population and Global Health, The University of Melbourne, Parkville, VIC 3010, Australia

**Keywords:** influenza viral dynamics, cell-surface mucin MUC1, immune response, mathematical models

## Abstract

MUC1 belongs to the family of cell surface (cs-) mucins. Experimental evidence indicates that its presence reduces in vivo influenza viral infection severity. However, the mechanisms by which MUC1 influences viral dynamics and the host immune response are not yet well understood, limiting our ability to predict the efficacy of potential treatments that target MUC1. To address this limitation, we use available in vivo kinetic data for both virus and macrophage populations in wildtype and MUC1 knockout mice. We apply two mathematical models of within-host influenza dynamics to this data. The models differ in how they categorise the mechanisms of viral control. Both models provide evidence that MUC1 reduces the susceptibility of epithelial cells to influenza virus and regulates macrophage recruitment. Furthermore, we predict and compare some key infection-related quantities between the two mice groups. We find that MUC1 significantly reduces the basic reproduction number of viral replication as well as the number of cumulative macrophages but has little impact on the cumulative viral load. Our analyses suggest that the viral replication rate in the early stages of infection influences the kinetics of the host immune response, with consequences for infection outcomes, such as severity. We also show that MUC1 plays a strong anti-inflammatory role in the regulation of the host immune response. This study improves our understanding of the dynamic role of MUC1 against influenza infection and may support the development of novel antiviral treatments and immunomodulators that target MUC1.

## 1. Introduction

Influenza is a contagious respiratory disease. It remains a major public health burden that affects and threatens millions of people each year [1]. Influenza virus (IV) primarily attacks the epithelial cells that line the upper respiratory tract (URT) of the host, causing an acute infection [2]. The host immune response has been shown to play an important role against influenza infection [3,4]. As part of the innate immune response, macrophages that reside in airways limit viral dissemination through phagocytosis of viral particles and prevent the virus from spreading to the lungs [5,6]. Activated macrophages produce inflammatory molecules, such as TNF-α, which stimulates recruitment of additional immune cells, such as monocyte-derived macrophages (MDMs) to the site of infection. These molecules also facilitate the activation of adaptive immune responses, such as maturation of B cells and effector CD8${}^{+}$ T cells [7]. Thus, macrophages play a critical role against influenza viral infection [8,9,10].

However, recruited macrophages also amplify inflammation. Overstimulation of the host immune response can lead to pathology, indicating that there is a subtle balance between a protective and a destructive response [1,11]. A dysregulated immune response, often marked by an excessive recruitment of macrophages to the site of infection and a high level of cytokine production, can lead to lung pathology, causing serious and sometimes fatal infection outcomes [12,13,14,15].

MUC1 belongs to the family of cell surface (cs-) mucin and is constitutively expressed at the surface of respiratory epithelial cells and macrophages, as reviewed in [16,17,18]. It was shown to be capable of modulating cytokine production in vitro viral infection [19,20,21] and in vivo bacterial infection [22,23]. More recently, McAuley and colleagues investigated the in vivo effects of MUC1 on influenza viral infection [24]. They first intranasally infected wildtype (WT) and MUC1-knockout (KO) mice with influenza A virus, then measured and compared the kinetic time-series data of viral load as well as different immune cells between the two groups. They found that the virus grows more quickly and reaches a peak earlier in MUC1-KO mice. Mice displayed a more enhanced inflammatory response, dominated by a higher number of macrophages and a high level of cytokine production. Based on these observations, they hypothesized that MUC1 acts as physical barrier to prevent virus from infecting epithelial cells and contribute to regulation of the host immune response. However, the potential effects of MUC1 in vivo are poorly quantified, limiting our ability to predict the efficacy of potential treatments that target MUC1. To address this limitation, we incorporated the hypothesized effects of MUC1 into mathematical models of influenza viral dynamics and applied Bayesian inference to estimate key parameter values and provided new quantitative insight into the role of MUC1 in shaping influenza virus infection and the host immune response.

Influenza viral dynamics models have been used to study many aspects of influenza infection and the host immune response, as reviewed in [25]. Studies focusing on the immune system have used viral dynamics models to study various types of immunological data, sharpening to our understanding of the contribution of different immunological components to influenza viral infection [26,27,28].

In this work, we use available in vivo kinetic data for both virus and macrophage populations in wildtype and MUC1 knockout mice. We analyze the data with two mathematical models of influenza viral dynamics under a Bayesian framework, quantifying the potential effects of cs-mucin MUC1 in influenza infection. The two models differ in how they categorise mechanisms of viral control. We also use the data-calibrated models to evaluate and analyze the dependence of various infection-related quantities on MUC1 expression. Finally, we discuss the biological implications of our results.

## 2. Results

### 2.1. Model Fitting

In vivo viral load and macrophage data in WT and MUC1-KO mice were used in model fitting. We fitted a Target cell-Infected-cell-Virus (TIV) model (Equations (Equation 1)–(4) in Materials and Methods) and an Immune Response (IR) model (Equations (Equation 5)–(18)) to the data, respectively. MUC1 has been suggested to prevent virus from infecting epithelial cells. It also has been implicated in the regulation of the host innate immune response, associated with macrophage recruitment [24]. As detailed in Materials and Methods, both models capture these effects. The reduction in susceptibility of target cells is captured by a parameter $\varepsilon_{1}$, modulating viral infectivity to the target cells in $dT/dt=(1-\varepsilon_{1})\beta TV$ (Equation (Equation 1)). The effect of the limitation of macrophage recruitment is captured by a parameter $\varepsilon_{2}$ and is modelled in $dM/dt=s+\left(1-\varepsilon_{2}\right)\phi I-\delta_{M}M$ (Equation (4)). In the absence of MUC1 expression, e.g., in MUC1-KO mice, we set $\varepsilon_{1}=\varepsilon_{2}=0$ to represent a complete knockout effect.

The fitting results are shown in Figure 1. The median of the posterior prediction (solid line) and a 95% predict interval (PI, shaded area) were computed from 4000 model simulation based on 4000 samples from the posterior distribution of model parameters (provided in Appendix A). The trend for both the viral kinetics (Figure 1A,B) and macrophages dynamics (Figure 1C,D) is well captured by the median prediction in both models, suggesting that both models are able to explain the data. Moreover, the narrow 95% PI indicates a relatively high certainty level for model predictions.

### 2.2. Estimates of MUC1 Parameters

The marginal posterior densities for $\varepsilon_{1}$ and $\varepsilon_{2}$ provide insight into the role of MUC1. The median parameter estimates and their associated 95% credible intervals (CIs) are given in Table 1. The median estimate of MUC1 on reduction of viral infectivity ($\varepsilon_{1}$) is 0.44 (95% CI: 0.23–0.71) in the TIV model and 0.42 (95% CI: 0.22–0.58) in the IR model. Furthermore, the estimated median values of MUC1 on regulation of macrophage recruitment ($\varepsilon_{2}$) are 0.45 (95% CI: 0.18–0.64) and 0.38 (95% CI: 0.06–0.63) in the TIV and IR models, respectively. Biologically, the median estimate for $\varepsilon_{1}$ indicates that the presence of MUC1 reduces the rate of virus infection to epithelial cells by 44% (for the TIV model) or 42% (for the IR model). In addition, the median estimate for $\varepsilon_{2}$ indicates that the presence of MUC1 reduces the recruitment rate of macrophages induced by infected cells by 45% ( for the TIV model) or 38% (for the IR model).

The posterior-median estimates are qualitatively consistent between the two models. The median estimates of $\varepsilon_{1}$ and $\varepsilon_{2}$ both exclude 0 (in the 95% credible interval) for the WT group (for both the TIV and IR models), indicating a reduced viral infectivity (i.e., $(1-\varepsilon_{1})\beta$) and a reduced rate of macrophage recruitment induced by infected cells (i.e., $(1-\varepsilon_{2})\phi$) in the presence of MUC1. The results support the experimental hypothesis [24] and provide quantitative evidence that the presence of MUC1 reduces viral infectivity to epithelial cells. They also provide evidence that MUC1 reduces macrophage recruitment and thus regulates the host innate immune response.

Detailed posteriors of model parameters are provided in Appendix A, and correlation maps of the estimated parameters for the TIV and IR models are given in Appendix A. There is a low correlation coefficient between $\varepsilon_{1}$ and $\varepsilon_{2}$ for the TIV (R=0.08) and IR (R=−0.22) models, suggesting the two parameters have a weak relationship. In particular, we found that the posterior-median estimate of the phagocytosis rate of virus by macrophages ($\kappa_{M}$) is approximately $10^{-8}$ for the TIV model, and the estimate is in agreement with the estimate for the IR model (Appendix A). We used the median estimates of model parameters to compute the ratio of macrophage-mediated viral decay ($\kappa_{M}M\left(t\right)$) to overall viral decay rate in the TIV ($\kappa_{M}M\left(t\right)+\delta_{V}$) and IR ($\kappa_{M}M\left(t\right)+\delta_{V}+\kappa_{AS}A_{S}\left(t\right)+\kappa_{AL}A_{L}\left(t\right)$) models as a time-series, respectively. We found that $\kappa_{M}M\left(t\right)$ only has a minor contribution to viral clearance (Appendix A). The result suggests that macrophages, although important to maintain gas exchange in lungs and reduce infection severity, are not directly involved in limiting viral replication, as shown in [29,30].

### 2.3. Prediction of Infection-Related Quantities

Influenza pathogenesis is often associated with a high viral load and an overstimulated immune response [15]. In the absence of MUC1, mice showed a significantly high mortality rate [24]. Here, we use the 4000 joint posterior distributions to predict the impact of MUC1 on some key infection-related quantities that likely influence infection severity. We then compare these quantities between the two models.

The basic reproduction number of viral replication ($R_{0}$) is defined as the average number of secondary infected cells that are produced by an initially infected cell when the target cell population is not depleted and is fully susceptible [31]. An infection may be established only if $R_{0}>1$. It is a critical indicator that quantifies how fast an infection is established and evolved.

Figure 2A,B show the $R_{0}$ between WT and MUC1-KO groups in the TIV and IR models, respectively. Both models predict a significantly higher median value of $R_{0}$ (dashed line) in the MUC1-KO group (20 in MUC1-KO group versus 11.1 in WT group for the TIV model, and 45.6 versus 26.4 for the IR model). The estimates of $R_{0}$ are comparable to previous estimates from fitting viral dynamics models to viral kinetic data in humans [32] and mice [33].

To assess the impact of MUC1 on viral dynamics, we compute the area under the viral load (without log-transformation) curve, which is often used as a marker for infectiousness (shown in Equation (Equation 21) in Materials and Methods). Both the TIV (Figure 2C) and IR (Figure 2D) models predict very similar $\log_{10}$(AUC${}_{V}$) in WT and MUC1-KO mice. This implies that a paucity of MUC1 expression has little, if any, effect on the cumulative viral load. This observation is supported by data in [24] in which they found that MUC1-KO mice were still capable of clearing virus after day 7 post infection.

An excessive accumulation of macrophages is considered to be a hallmark for severe infection, often observed in highly pathogenic influenza viral infection [14]. We use the area under the macrophage time-series curve (without log-transformation; Equation (22) in Materials and Methods) as a surrogate for the strength of immune response stimulation and explore the dependence of the AUC${}_{M}$ on MUC1. As shown in Figure 2E,F, both models predict a higher median value of $\log_{10}$(AUC${}_{M}$) in MUC1-KO mice compared to WT mice. This suggests that MUC1 reduces the accumulation of macrophages and thus contributes to the regulation of the host immune response.

We also assessed the influence of MUC1 on peak viral load (Appendix A) and peak viral load time (Appendix A) for the two models. Both models predict that the presence of MUC1 delays the time at which viral load peaks but only has a subtle influence on the magnitude of peak viral load, as shown in [24].

In summary, both models predict a higher value of $R_{0}$ (Figure 2A,B) and increased macrophage accumulation (Figure 2E,F) in the absence of MUC1 expression. The results emphasise the dual roles for MUC1 in reducing viral infectivity and limiting macrophage recruitment. Furthermore, they suggest that the absence of MUC1, while not driving a significant increase in cumulative viral load, facilitates viral replication and dissemination within the host in the early stages of infection. More epithelial cells are infected in a short time interval, heightening macrophage recruitment, likely contributing to lung pathology and providing an explanation for the heightened mortality rate in MUC1 KO mice.

### 2.4. Delineation the Effects of MUC1 on Macrophage Recruitment

We showed that the presence of MUC1 reduces AUC${}_{M}$ (Figure 2E,F), which may alleviate infection severity. The accumulation of macrophages is not only directly impacted by the regulatory effect of MUC1, (i.e., $\varepsilon_{2}$), but is also indirectly affected by antigen levels, which are influenced by $\varepsilon_{1}$ through modulating dynamics for infected cells (*I*). Here, we analyze the relative contribution of the two effects of MUC1 on the AUC${}_{M}$. We use the macrophage reduction efficiency, defined as the decrease in the area under the macrophage curve in wild type mice (AUC${}_{M,WT}$) relative to the AUC of the macrophage curve in MUC1 knockout mice (AUC${}_{M,KO}$):$$\begin{array}{c} \mathrm{Macrophage}\;\mathrm{Reduction}\;\mathrm{Efficiency}=1-\frac{AUC_{M,WT}}{AUC_{M,KO}} \end{array}$$

Figure 3 shows the estimated marginal posterior density of $\varepsilon_{1}$ and $\varepsilon_{2}$ for the TIV model (top panel) and a heatmap of the dependence of macrophage reduction efficiency on $\varepsilon_{1}$ and $\varepsilon_{2}$ (bottom panel). The heatmap predicts the dependence of the macrophage reduction efficiency for various values of $\varepsilon_{1}$ and $\varepsilon_{2}$ within the 95% CI. We observe that a higher $\varepsilon_{1}$ or $\varepsilon_{2}$ leads to a higher macrophage reduction level, suggesting that both effects contribute to reduce the accumulation of macrophages. However, the macrophage reduction efficiency is notably more sensitive to changes in $\varepsilon_{2}$. In particular, taking the median parameter values as a reasonable prediction point (black circle), the rate of change in the macrophage reduction efficiency is strongly dependent on $\varepsilon_{2}$ and only weakly dependent on $\varepsilon_{1}$. (indicated by the arrow line). The result suggests that the reduction in macrophage recruitment (i.e., via an increase in $\varepsilon_{2}$) is more influential than the reduction of infected cells (i.e., via a reduction in $\varepsilon_{1}$) in determining AUC${}_{M}$. We also assess the macrophage reduction efficiency as a function of $\varepsilon_{1}$ and $\varepsilon_{2}$ for the IR model. As shown in Figure 4, the results are qualitatively consistent—the macrophage reduction efficiency is strongly influenced by $\varepsilon_{2}$.

Both models predict a strong effect for $\varepsilon_{2}$ and relatively small effect for $\varepsilon_{1}$ on the AUC${}_{M}$. This is understood by recalling that the presence of MUC1 does not significantly influence the cumulative viral load, as shown in Figure 2C,D. Thus, a change in the reduction of viral infectivity to target cells ($\varepsilon_{1}$) has only a minor effect on the AUC${}_{M}$. The results emphasise a strong regulatory effect of MUC1 on macrophage accumulation.

## 3. Discussion

In this work, we studied the in vivo immunological effects of cs-mucin MUC1 in influenza viral infection. To the best of our knowledge, this is the first study to incorporate the dynamical roles of MUC1 into models of influenza virus dynamics. In our models, MUC1 regulates both viral replication and macrophage recruitment. We incorporated the experimentally hypothesized roles of MUC1 into two mathematical models and fitted kinetic data of both virus and macrophage populations to the models in a Bayesian framework. Our estimation results (Table 1) provide evidence that MUC1 reduces the susceptibility of epithelial cells to viral infection. They also provide evidence that MUC1 limits the recruitment of macrophages and thus regulates the host immune response. Both models predict the influence of MUC1 on various infection-related quantities (Figure 2). While the expression of MUC1 has little impact on the cumulative viral load (AUC${}_{V}$), it delays viral infection by reducing the basic reproduction number of viral replication (Figure 2A,B) and delaying viral load peak time (Appendix A). More importantly, we found that the presence of MUC1 significantly reduces the accumulation of macrophages (Figure 2E,F). The decreased level of macrophages is primarily driven by the direct regulatory effect ($\varepsilon_{2}$) of MUC1 on macrophage recruitment (Figure 3 and Figure 4).

Our model-based analyses provide new insight into the mechanisms by which MUC1 influences viral dynamics and the host immune response. This is also the first study that we are aware of that provides quantitative estimates of the in vivo effects of cs-mucin MUC1 on influenza infection. Our analyses enhance our ability to predict the efficacy of potential treatments that target MUC1. Influenza pathogenesis is often marked by a high viral load, and infection of epithelial cells is a key determinant of the level of viral load [11,15,34]. MUC1 is rapidly stimulated at the surface of epithelial cells and macrophages on infection, and is thought to act as a “releasable decoy”, preventing virus from attaching and infecting the cells, thereby reducing viral infectivity [17]. Regardless of the specific mechanism, our model predictions suggest that MUC1 only effectively reduce $R_{0}$ (Figure 2B) but not the AUC${}_{V}$ (Figure 2C,D). The biological implications of this are two-fold. Firstly, MUC1, as part of the innate immune response, has been shown to be rapidly up-regulated within a few hours post in vitro infection [21]. The decreased $R_{0}$ suggests that MUC1 expression contributes to limit and delay viral infection, and more importantly, to prevent viral dissemination within the host. This provides strong protection to the host and reduce infection severity. Viral spread to the lower respiratory tract (LRT) is known to cause complications, leading to more severe infection outcomes [34]. Secondly, the comparable AUC${}_{V}$ between WT and MUC1-KO group implies that a lack of cs-mucin MUC1 protection have a subtle influence on other immunological components that are responsible for viral clearance, such as the host adaptive immune response. This may be partially supported in [24], where MUC1-KO mice were shown to clear virus from the lungs at day 7 post infection. A more comprehensive dataset that captures the dynamics of antibodies or effector CD8${}^{+}$ T cells would greatly improve our understanding of the impact on MUC1 to the adaptive immune response.

Beyond these virological indicators, viral pathogenesis is also associated with the strength of the host immune response induced by influenza infection. An excessive recruitment of macrophages to the site of infection is a hallmark of overstimulated immune responses [14,34]. The anti-inflammatory role of MUC1 has been shown to inhibit activation of Toll-like receptors (TLRs) in macrophages and infected cells [24]. In MUC1-KO mice, our models predicted a significantly enhanced level of AUC${}_{M}$ (Figure 2E,F), which may reflect the high mortality rate in the group. This finding emphasises the importance of quantities related to the immune response, which can be critical indicators for predicting the severity of infection and facilitating the assessment of antiviral therapies, as suggested in [34,35]. Furthermore, we showed that the decreased AUC${}_{M}$ is primarily due to the direct regulatory effect of MUC1 on macrophages (i.e., $\varepsilon_{2}$), which highlights a strong anti-inflammatory effect for MUC1. This may support the development of novel immunomodulators that target cs-mucin MUC1.

In conducting this study, we applied two mathematical models to the kinetic data. The models models differ in how they model adaptive immunity. We compared the key estimation results of MUC1’s effects and model predictions of infection-related quantities between the two models. We found that both models fit the in vivo viral load and macrophage data well (Figure 1), giving comparable parameter estimates and consistent biological insights.

One of the most important applications of viral dynamic models is to estimate key kinetic parameters, as reviewed in [25]. Model selection for data fitting is an important but unresolved challenge in influenza dynamics modelling due to limited time-series data on numerous quantities of interested. Parameter estimates vary substantially between different studies, and the predictive power of any given model is influenced by the selection of model components, as showed in previous work by our group [27,35] and others [36]. In our study, there are advantages and disadvantages in applying the TIV and IR models. Due to its simple model structure, the TIV model is more computationally efficient. However, its lack of a detailed characterization of adaptive immunity makes the model difficult to use to explore potential interactions between different immunological components, e.g., interactions between macrophages and CD8${}^{+}$ T cells. The IR model, on the other hand, is more computationally intensive and has far more parameters to either estimate or determine from the literature. However, it is more suitable for explaining in vivo kinetic viral load data to which adaptive immunity has been shown to have an influence. It also provides a platform to study more complicated virus-immunity dynamics and interaction between different components of immune responses.

Neither the TIV nor IR models consider the full spectrum of host immune response which are known to contribute to viral control and that have been included in other modelling works, e.g., interferon dynamics [27,28]. Regardless, we argue our two models are sufficient for this study in which we focus on the influence of MUC1 on viral dynamics and macrophage kinetics, which are both explicitly considered in the models. Furthermore, there is no evidence to suggest that MUC1 has an impact on the adaptive immune response. Combined with the observation that MUC1-KO mice clear virus after day 7 post infection [24], the effects of MUC1 may be minimally influenced by the detailed dynamics of adaptive immunity.

Our study has some limitations. We only incorporated the two hypothesized effects of cs-mucin MUC1 on influenza viral infection into our mathematical models, but did not consider the detailed dynamics of MUC1 itself due to a lack of MUC1 kinetic data. As a result, the critical timing at which MUC1 starts to take effect has not been estimated. This could be an important factor that influences disease severity [17]. In future work, explicitly modelling the time dependent MUC1 effects would be of interest given availability of time-series data of MUC1 expression. Another limitation is that we assumed a fixed adaptive immune response, such that the adaptive immune responses dominate viral clearance at day 5 post infection regardless of MUC1 expression [27,37]. Though there is no evidence so far that MUC1 would affect the magnitude and/or timing of the adaptive immune response, extension of the IR model to allow for such an effect may be of interest.

## 4. Materials and Methods

### 4.1. Mathematical Models

In this study, we considered two mathematical models that are often used to study within-host influenza dynamics, but which differ in how they categorise the mechanisms of viral control.

#### 4.1.1. The TIV Model

The Target cell-Infected cell-Virus (TIV) model depicts a simple but fundamental interaction between target cells and influenza virus, as originally presented in [32]. To estimate the in vivo impacts of MUC1, we incorporate the two hypothesized effects of MUC1 on viral infectivity and innate immune responses into the TIV model. We also consider a component of macrophage dynamics and critical interactions between macrophages and virus. The model is described by a set of ordinary differential equations (ODEs): (1)$$\begin{array}{cc} \; & \frac{dT}{dt}=gT\left(1-\frac{T+I}{T_{max}}\right)-\left(1-\varepsilon_{1}\right)\beta TV, \end{array}$$
(2)$$\begin{array}{cc} \; & \frac{dI}{dt}=\left(1-\varepsilon_{1}\right)\beta TV-\delta_{I}I, \end{array}$$(3)$$\begin{array}{cc} \; & \frac{dV}{dt}=pI-\delta_{V}V-\kappa_{M}MV, \end{array}$$(4)$$\begin{array}{cc} \; & \frac{dM}{dt}=s+\left(1-\varepsilon_{2}\right)\phi I-\delta_{M}M. \end{array}$$

Equations (Equation 1)–(3) describe the interaction between virus and epithelial cells. In detail, epithelial cells (*T*), the target cells for influenza virus, are infected with virus (*V*) and become infected cells (*I*) at an infectivity rate βV per day. Target cells are replenished at a rate $gT(1-\left(T+I\right)/T_{max})$, where $T_{max}$ is the maximal number of epithelial cells that line the upper respiratory tract (URT). The infectivity rate is modified by MUC1, parameterised by $\varepsilon_{1}$. Infected cells produce free virus at a rate *p* per day. Apoptosis occurs at a rate $\delta_{I}$ per day. We do not explicitly model the role of macrophages in removing apoptotic infected cells [3,6,38]. While an established role of macrophages, it is not required as we have no data on dead cell dynamics and so our model does not include the dynamics of the dead cell population. The decrease of free virus is either due to natural decay at a constant rate $\delta_{V}$ per day, or internalization by macrophages (*M*) at a rate $\kappa_{M}M$.

Equation (4) models the dynamics of macrophages. We assume a constant supplementary rate and a decay rate of macrophages at *s* and $\delta_{M}$ per day, respectively. Upon infection, monocytes are recruited from peripheral blood to the site of infection and become monocyte-derived macrophages (MDMs) in the presence of cytokines. We assume the recruitment rate is proportional to the level of infected cells, ϕI, as infected cells contribute to cytokines production. The cs-mucin MUC1 regulates the recruitment rate of macrophages, parameterised by $\varepsilon_{2}$.

#### 4.1.2. The IR Model

The immune response (IR) model is based on the TIV model and includes a detailed adaptive immune response, which contributes to viral clearance over a distinct timescale [28]. The model is formulated by a system of ODEs: (5)$$\begin{array}{cc} \; & \frac{dT}{dt}=gT\left(1-\frac{T+I}{T_{max}}\right)-\left(1-\varepsilon_{1}\right)\beta TV, \end{array}$$(6)$$\begin{array}{cc} \; & \frac{dI}{dt}=\left(1-\varepsilon_{1}\right)\beta TV-\delta_{I}I-\kappa_{E}EI, \end{array}$$(7)$$\begin{array}{cc} \; & \frac{dV}{dt}=pI-\delta_{V}V-\kappa_{M}MV-\kappa_{AS}A_{S}V-\kappa_{AL}A_{L}V, \end{array}$$(8)$$\begin{array}{cc} \; & \frac{dM}{dt}=s+\left(1-\varepsilon_{2}\right)\phi I-\delta_{M}M, \end{array}$$(9)$$\begin{array}{cc} \; & \frac{dE_{0}}{dt}=-\gamma_{E}\frac{V}{V+E_{50}}E_{0}, \end{array}$$(10)$$\begin{array}{cc} \; & \frac{dE_{1}}{dt}=\gamma_{E}\frac{V}{V+E_{50}}E_{0}-\frac{n_{E}}{\tau_{E}}E_{1}, \end{array}$$(11)$$\begin{array}{cc} \; & \frac{dE_{i}}{dt}=\frac{n_{E}}{\tau_{E}}\left(E_{i-1}-E_{i}\right),\;i=2,\ldots ,n_{E} \end{array}$$(12)$$\begin{array}{cc} \; & \frac{dE}{dt}=\phi_{E}\frac{n_{E}}{\tau_{E}}E_{n_{E}}-\delta_{E}E, \end{array}$$(13)$$\begin{array}{cc} \; & \frac{dB_{0}}{dt}=-\gamma_{B}\frac{V}{V+B_{50}}B_{0}, \end{array}$$(14)$$\begin{array}{cc} \; & \frac{dB_{1}}{dt}=\gamma_{B}\frac{V}{V+B_{50}}B_{0}-\frac{n_{B}}{\tau_{B}}B_{1}, \end{array}$$(15)$$\begin{array}{cc} \; & \frac{dB_{i}}{dt}=\frac{n_{B}}{\tau_{B}}\left(B_{i-1}-B_{i}\right),\;i=2,\ldots ,n_{B} \end{array}$$(16)$$\begin{array}{cc} \; & \frac{dP}{dt}=\phi_{p}\frac{n_{B}}{\tau_{B}}B_{n_{B}}-\delta_{p}P, \end{array}$$(17)$$\begin{array}{cc} \; & \frac{dA_{S}}{dt}=\mu_{S}P-\delta_{AS}A_{S}, \end{array}$$(18)$$\begin{array}{cc} \; & \frac{dA_{L}}{dt}=\mu_{L}P-\delta_{AL}A_{L}. \end{array}$$

Equations (Equation 5)–(8) retain the skeleton of the TIV model, describing the essential target cell-virus dynamics, except for additional components in dI/dt and dV/dt related to adaptive immune responses. $\kappa_{E}E$ in Equation (6) represents the rate of infected cells lysis by effector CD8${}^{+}$ T cells. The extra terms $\kappa_{AS}A_{S}$ and $\kappa_{LS}A_{L}$ in Equation (7) represent virus clearance mediated by a short-lived ($A_{S}$, e.g., IgM) and a long-lasting antibody ($A_{L}$, e.g., IgG), respectively.

Equations (9)–(12) describe a major component of the cellular adaptive immune response mediated by CD8${}^{+}$ T cells. Naïve CD8${}^{+}$ T cells ($E_{0}$) initiate proliferation and differentiate into effector cells $E_{1}$ on stimulation via antigen-presentation at a rate $\gamma_{E}V/\left(V+E_{50}\right)$, where $\gamma_{E}$ is the maximal stimulation rate, and $E_{50}$ is a half saturation level at which half of the stimulation rate is obtained (as shown in Equation (9)). Effector cells $E_{1}$ perform programmed proliferation to $E_{i}$ where *i* denotes proliferation stages (Equations (10) and (11)) for $\tau_{E}$ days, experience through $n_{E}$ stages [39], finally becoming mature effector cytotoxic T lymphocytes (*E*) at a rate $\phi_{E}$ at the final stage. The decay rate of *E* is $\delta_{E}$, as shown in Equation (12).

Similarly, the dynamics of the humoral adaptive immune response are described by Equations (13)–(16). Naïve B cells ($B_{0}$) start to proliferate and differentiate into plasma cells ($B_{1}$) once stimulated by virus at a rate $\gamma_{B}V/\left(V+B_{50}\right)$, where $\gamma_{B}$ is the maximal stimulation rate and $B_{50}$ is a half-saturation level, as shown in Equation (13). Equations (14) and (15) capture how plasma cells ($B_{1}$) undergo programmed proliferation through $n_{B}$ stages into $B_{i}$, where *i* denotes proliferation stages, for $\tau_{B}$ days [39]. Finally, mature plasma cells *P* (Equation (16)) are produced at a rate $\phi_{B}$ and decay at a rate $\delta_{p}$.

Equations (17) and (18) describe the dynamics of a short-lived antibody ($A_{S}$) and a long-lived antibody ($A_{L}$). $A_{S}$ and $A_{L}$ are produced by plasma cells (*P*) at rates $\mu_{S}$ and $\mu_{L}$ and decay at rates $\delta_{AS}$ and $\delta_{AL}$, respectively.

### 4.2. Statistical Inference

We extracted the kinetic data of both virus and macrophage population in wild type (WT) and MUC1 knockout mice using WebPlotDigitizer (version 4.4) from [24]. In the study, groups of wild type and MUC1-KO mice were intranasally infected with influenza A virus (PR8). There were five mice in each group. We assumed the variability of virus and macrophage data between different mice within the same group was due to measurement error, so that the data from different mice were pooled together for analysis.

We took a Bayesian inference approach to fit the TIV and IR model (detailed in Model) to the log-transformed kinetic data. In detail, our model has 10 parameters to estimate, and the parameter space is denoted as $\Phi =(\varepsilon_{1},\beta ,\delta_{I},p,\delta_{V},s,\delta_{M},\varepsilon_{2},\kappa_{M},\phi )$. Upon calibrating the IR model, we fixed all parameters of the adaptive immune responses (e.g., all parameters in Equations (9)–(18)) to previous estimated values in the literature [27,39]. We fixed the parameters because estimating the immunological effects of adaptive immunity is not a focus of this study, and [24] does not provide sufficient data for estimation of these parameters. We chose the number of effector T cell and B cell division cycle (i.e., $n_{E}$ and $n_{B}$) and the total proliferation time of the cells (i.e., $\tau_{E}$ and $\tau_{B}$) from [39], such that the adaptive immune responses only become activated five days post infection. The fixed parameter values are given in Appendix A.

Furthermore, we assumed WT and MUC1-KO mice only differ in $\varepsilon_{1}$ and $\varepsilon_{2}$, a reasonable assumption given inbred mice and use of the same virus for all experiment. We fitted log-transformed WT and MUC1-KO data simultaneously to the models with the same parameter vector set, only differing except for $\varepsilon_{1}$ and $\varepsilon_{2}$, which were set to $\varepsilon_{1}=\varepsilon_{2}=0$ for MUC1-KO mice. The prior distribution for model parameters (Φ) is given in Appendix A. The distribution of the observed log-transformed viral load and macrophage measurement is assumed to be a normal distribution with a mean value given by the model simulation results and standard deviation (SD) parameter with prior distribution of a normal distribution with a mean of 0 and a SD of 1.

Model fitting was performed in R (version 4.0.2) and Stan (Rstan 2.21.0). Hamiltonian Monte Carlo (HMC) optimized by the No-U-Turn Sampler (NUTS) [40] was implemented to draw samples from the joint posterior distribution of the model parameters. A detailed theoretical foundation of HMC can be found in [41]. In particular, we used four chains with different starting points and ran 2000 iterations for each chain, discarding the first 1000 iterations as burn-in. We retained 4000 samples in total from 4 chains (1000 for each) after the burn-in iterations. The marginal posterior and prior density for all parameters are shown in Appendix A. We calculated the median and quantiles of 2.5% and 97.5% of the 4000 model outputs at each time for posterior prediction and a 95% prediction interval (PI), respectively (e.g., Figure 2).

### 4.3. Infection-Related Quantities

The basic reproduction number of viral replication ($R_{0}$) is given by
(19)$$\begin{array}{c} R_{0}=\frac{\left(1-\varepsilon_{1}\right)\beta T_{0}V}{\left(\delta_{I}+\kappa_{E}E\left(0\right)\right)\left(\delta_{V}+\kappa_{M}M_{0}+\kappa_{AS}A_{S}\left(0\right)+\kappa_{AL}A_{L}\left(0\right)\right)}, \end{array}$$
where $T_{0}$ is the initial number of epithelial cells, and $M_{0}$ is the number of macrophages in a disease-free equilibrium, given by $s/\delta_{M}$. $E\left(0\right),A_{S}\left(0\right)$ and $A_{L}\left(0\right)$ are initial values of effector CD8${}^{+}$ T cells, a short-term antibody and a long-term antibody, respectively, which are set to zero at the beginning of infection as mice are naive (i.e., not previously exposed to influenza). Therefore, $R_{0}$ simplifies to
(20)$$\begin{array}{c} R_{0}=\frac{\left(1-\varepsilon_{1}\right)\beta T_{0}V}{\delta_{I}\left(\delta_{V}+\kappa_{M}M_{0}\right)}. \end{array}$$

Please note that $\varepsilon_{1}=0$ in MUC1-KO group. The area under the viral load time-series curve (AUC${}_{V}$) and under the macrophage time-series curve (AUC${}_{M}$) are given by
(21)$$\begin{array}{cc} \; & AUC_{V}=\int_{0}^{\tau }V\left(t\right)dt, \end{array}$$
(22)$$\begin{array}{cc} \; & AUC_{M}=\int_{0}^{\tau }M\left(t\right)dt, \end{array}$$
where τ is a cut-off day for calculation. We set τ=14, which covers the duration of viral infection, macrophage dynamics and clinical dynamics in [24]. V(t) and M(t) are simulated time series of viral load and macrophages, respectively.

The estimates of the infection-related quantities were computed using the 4000 posterior samples by solving the ode solver ode15s in MATLAB R2019b with a relative tolerance of $1\times 10^{-5}$ and an absolute tolerance of $1\times 10^{-10}$. The initial values for different model components in the TIV model is $\left(T,I,V,M\right)=\left(1\times 10^{7},0,30,s/\delta_{M}\right)$, where *s* and $\delta_{M}$ are estimated from fitting the macrophage data to the model. For the IR model, the initial values were $\left(T,I,V,M,E_{0},E_{1}\cdots E,B_{0},B_{1}\cdots P,A_{S},A_{L}\right)=\left(1\times 10^{7},0,30,s/\delta_{M},100,0,\cdots 0,100,0,\cdots 0,0,0\right)$. The values of fixed parameters are given in Appendix A. All visualization was performed in R (version 4.0.2). Computer codes to produce all the figures in this study can be found at https://github.com/keli5734/MUC1 (accessed on 5 May 2021).

## Figures and Tables

**Figure 1 viruses-13-00850-f001:**
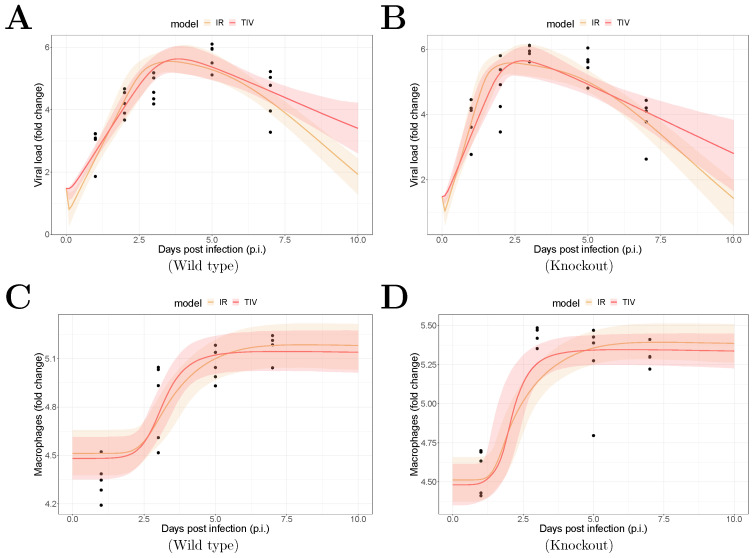
Results of model fitting for WT and MUC1-KO mice. Data are presented by solid circles. Panels (**A**,**B**) show the median of posterior predictions (solid line) and a 95% prediction interval (shaded area) of viral load data for both TIV (red) and IR (yellow) models for WT and MUC1-KO mice, respectively. Panels (**C**,**D**) show the model predictions of macrophage data in the two models for WT and MUC1-KO mice, respectively. The priors of model parameters are given in Appendix A. The posteriors of estimated model parameters are given in Appendix A.

**Figure 2 viruses-13-00850-f002:**
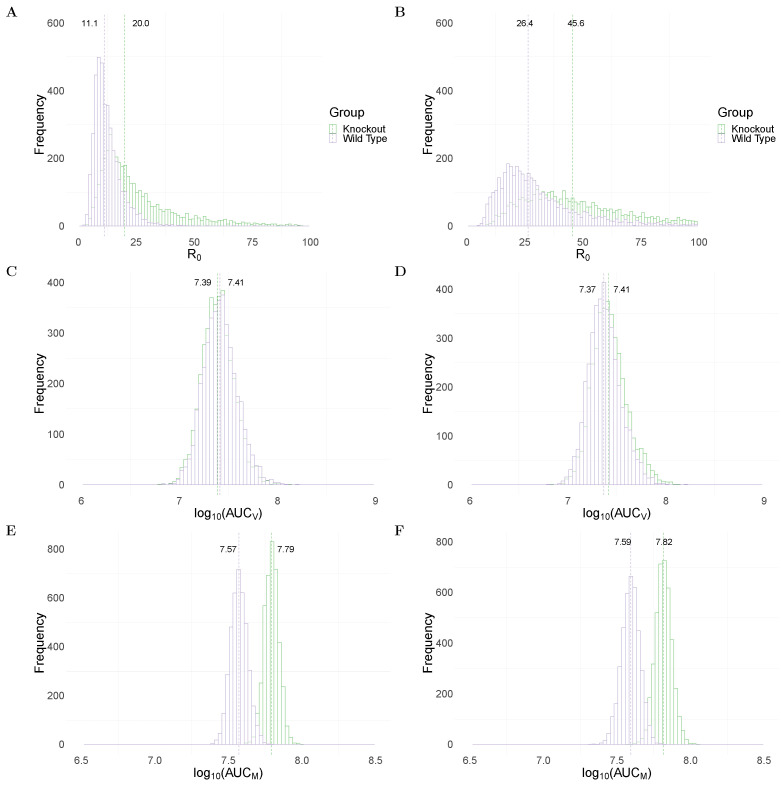
Comparison of model predictions for selected key biological quantities. Distributions are calculated using the 4000 joint posterior distributions. Panels (**A**,**B**) show the distribution of the basic reproduction number of viral replication in wildtype (purple) and MUC1-knockout (green) group in TIV (**left** panel) and IR models (**right** panel), respectively. Panels (**C**,**D**) show the distribution of the cumulative viral load in different mice groups in the two models. Panels (**E**,**F**) show the accumulative macrophages in WT and MUC1-KO mice group in the two models.

**Figure 3 viruses-13-00850-f003:**
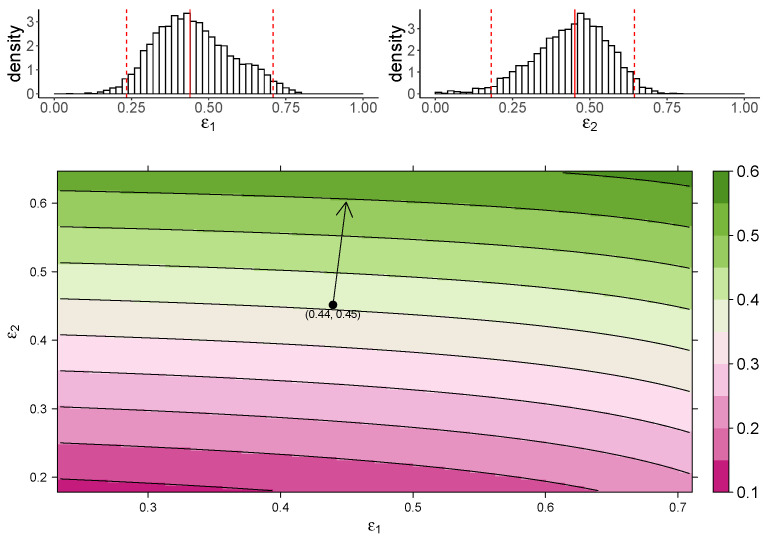
The dependence of the AUC${}_{M}$ on the effects of MUC1 for the TIV model. The upper panel shows the marginal posterior distribution of $\varepsilon_{1}$ (**left**) and $\varepsilon_{2}$ (**right**). The 95% credible interval (CI) for the parameters is indicated between the two red-dashed lines, and the red-solid line indicates parameters’ median value. The heatmap shows dependence of macrophage reduction efficiency on $\varepsilon_{1}$ and $\varepsilon_{2}$. The black circle indicates the pair of median values of $\varepsilon_{1}$ and $\varepsilon_{2}$, and the arrow indicates the direction of the rate of change in macrophage reduction efficiency at that point.

**Figure 4 viruses-13-00850-f004:**
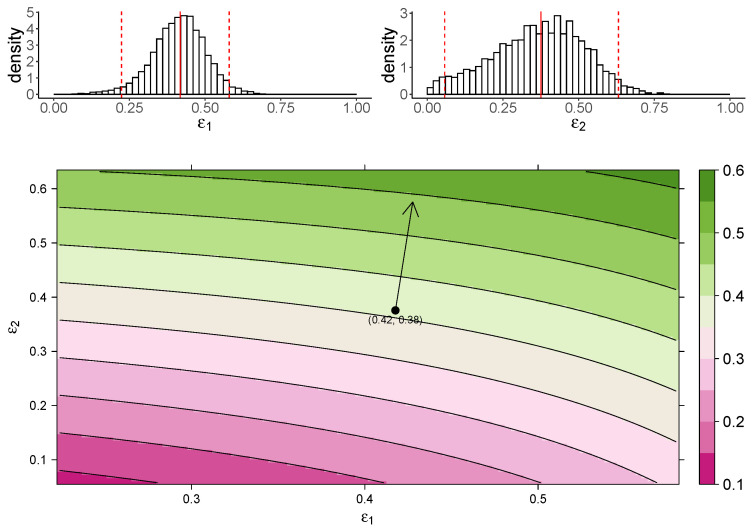
The dependence of the AUC${}_{M}$ on the effects of MUC1 for the IR model. The upper panel shows the marginal posterior distribution of $\varepsilon_{1}$ (**left**) and $\varepsilon_{2}$ (**right**). The 95% credible interval (CI) for the parameters is indicated between the two red-dashed lines, and the red-solid line indicates parameters’ median value. The heatmap shows dependence of macrophage reduction efficiency on $\varepsilon_{1}$ and $\varepsilon_{2}$. The black circle indicates the pair of median values of $\varepsilon_{1}$ and $\varepsilon_{2}$, and the arrow indicates the direction of the rate of change in macrophage reduction efficiency at that point.

**Table 1 viruses-13-00850-t001:** Estimates of MUC1 parameters and comparison between models. The estimates of MUC1 on reduction of target cell susceptibility to influenza virus ($\varepsilon_{1}$) and on reduction of macrophage recruitment rate induced by infected cells ($\varepsilon_{2}$). The lower and upper boundary of the 95% credible interval (CI) of the parameter is given by calculating the 2.5% and 97.5% quantile of the marginal posterior parameter distribution.

Parameter	Description	Median (95% CI)	
		**TIV**	**IR**
$\varepsilon_{1}$	The reduction in target cell susceptibility to infection due to MUC1	0.44 (0.23, 0.71)	0.42 (0.22, 0.58)
$\varepsilon_{2}$	The reduction in recruitment rate of macrophages due to MUC1	0.45 (0.18, 0.64)	0.38 (0.06, 0.63)

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
