# Peer review of "Modelling the Effect of MUC1 on Influenza Virus Infection Kinetics and Macrophage Dynamics"

_viruses, 2021, doi:10.3390/v13050850_

Round 1

Reviewer 1 Report

In this paper, the authors use two different viral dynamics models combined with macrophage recruitment and macrophage effects on infection, to understand the role of MUC1 (cell-surface mucin) on acute influenza infections. They use a Bayesian approach to estimate a subset of each models' parameters, and investigate two hypotheses about the specific actions of MUC1 during acute influenza infections, namely that MUC1 presents a physical barrier that prevents the infection of epithelial cells and that it contributes to the specific macrophage actions and thus the regulation of the host immune response, by studying MUC1 wild-type (WT) and knockout (KO) scenarios.

1) There seems to be little difference in observed viral loads between the WT and KO case in Figures 1A and 1B, yet ε1 is equal to or larger than ε2 in both models, and the authors conclude that "the presence of MUC1 reduces viral infectivity to epithelial cells," on the basis of the estimates for  ε1 and  ε2. What could explain the relative lack of change in peak viral loads if MUC1 is reducing infection of epithelial cells?

2) Lines 175-177: "This result suggests that the direct regulatory effect of MUC1 on macrophage recruitment has a dominant influence on the AUCM." Isn't this simply a consequence of how the effect of MUC1 was modelled, i.e. linearly through ε2?

3) In the TIV model, macrophages increase their population linearly through interactions with infected cells (the (1-ε2)φI) and reduce free virus, but have no direct killing effect on infected cells. Are inflammatory macrophages implicated in infected cell killing/removal, and what are the consequences of not including these mechanisms on both models' predictions? 

4) Should there be terms accounting for CD8+ T cell killing in the R0 for the IR model (Eq. 19)?

5) How were the number of proliferative compartments (Ei and Bi) determined?

6) In the IR model, what would be the consequences of assuming some pre-existing immunity/cross-reactivity (i.e. not setting memory T and B cells and antibodies to be 0 initially?

I also noted a few minor typos: 

-Lines 116-118: "The result suggests that macrophages, although is important to maintain gas exchange in lungs and reduce infection severity, is are not directly involved in limiting viral replication, as evidenced in [29,30]."

-Line 225: "Beyond these virological indicators"

Author Response

Dear reviewer, 

Thank you for the comments, and please see the response in the attachment. 

Kind regards,

Ke 

Reviewer 2 Report

Overall, this manuscript presents its interesting in silico findings in a rigorous way, with appropriate attention to both computational and virologic aspects. To make it even more convincing, I suggest the following. The main argumentation is based on the introduction of two parameters in the existing models, epsilon_1 and epsilon_2. It would be good

  •  to refer to instances in the literature where similar parameters, playing similar roles, have been used. Or to mention explicitly that this is a new idea, if that turns out to be the case.
  • to give a verbal quantitative interpretation of the estimates of these parameters. What does a reduction of epsilon_1 of 0.44 exactly mean biologically - does it simply mean that infectivity is half as much as with KO mice, or is it related to an increase of size one of the corresponding unit ? What precisely is the unit, if any, concentration or concentration/time or dimensionless ?
  • to present, if desired in the supplementary materials, the estimates obtained of these parameters when used for KO mice as well. This would greatly and quantitatively emphasize the importance of the effects observed in Wild type mice. Can the goodness-of-fit or R2 values or other indicators of the quality of fit be displayed in these models and the models already presented ?

Otherwise there seem to be some minor language typos on lines

  • 115 - 118
  • captions for Figure 3 and 4: The bold title of the figures seems gramatically wrong. The second non-bald sentence in the captions some words are apparently missing.  
  • 215

In the second sentence of 2.4 it could be pointed out that the influence of epsilon_1 is positive (before the word 'indirectly').

Author Response

(The authors gave the same response as above.)

Round 2

Reviewer 1 Report

The authors have answered all of my previous questions, thank you. The additions made in their revisions have all adequately clarified the previous points raised.